

# Predominance of Secondary Organic Aerosol to Particle-bound Reactive Oxygen Species Activity in Fine Ambient Aerosol

Jun Zhou[1,a], Miriam Elser[1,b], Ru-Jin Huang[1,2], Manuel Krapf[1], Roman Fröhlich[1], Deepika Bhattu[1], Giulia Stefenelli[1], Peter Zotter[4], Emily A. Bruns[1], Simone M. Pieber[1,c], Haiyan Ni[2], Qiyuan Wang[2], Yichen Wang[2], Yaqing Zhou[2], Chunying Chen[5], Mao Xiao[1], Jay G. Slowik[1], Samuel Brown[1,6], Laure-Estelle Cassagnes[1], Kaspar R. Daellenbach[1,d], Thomas Nussbaumer[4], Marianne Geiser[3], André S.H. Prévôt[1], Imad El-Haddad[1], Junji Cao[2], Urs Baltensperger[1], and Josef Dommen[1]

[1]Laboratory of Atmospheric Chemistry, Paul Scherrer Institute, 5232, Villigen, Switzerland

[2]Institute of Earth Environment, Chinese Academy of Sciences, Xi'an, 710061, China

[3]Institute of Anatomy, University of Bern, 3012, Bern, Switzerland

[4]Bioenergy Research Group, Engineering and Architecture, Lucerne University of Applied Sciences and Arts, 6048, Horw, Switzerland

[5]CAS Key Laboratory for Biological Effects of Nanomaterials and Nanosafety, National Centre for Nanoscience and Technology, Beijing 100191, China

[6]Institute for Atmospheric and Climate Science, ETH, 8092, Zurich, Switzerland

[a]now at: Graduate School of Global Environmental Studies, Kyoto University, Kyoto, 606-8501, Japan

[b]now at: Laboratory for Advanced Analytical Technologies, Empa, 8600 Dübendorf, Switzerland

[c]now at: Laboratory for Air Pollution/Environmental Technology, Empa, 8600 Dübendorf, Switzerland

[d]now at: Institute for Atmospheric and Earth System Research/Physics, Faculty of Science, University of Helsinki, 00014, Helsinki, Finland

*Correspondence to*: Josef Dommen (josef.dommen@psi.ch) and Ru-Jin Huang (rujin.huang@ieecas.cn)

**Abstract:** Reactive oxygen species (ROS) are believed to contribute to the adverse health effects of aerosols. This may happen by inhaled particle-bound (exogenic) ROS (PB-ROS) or by ROS formed within the respiratory tract by certain aerosol components (endogenic ROS). We investigated the chemical composition of aerosols and their exogenic ROS content at the two contrasting locations Beijing (China) and Bern (Switzerland). We apportioned the ambient organic aerosol to different sources and attributed the observed PB-ROS to them. The oxygenated organic aerosol (OOA, a proxy for secondary organic aerosol, SOA) explained the highest fraction of the exogenic ROS concentration variance at both locations. We also characterized primary and secondary aerosol emissions generated from different biogenic and anthropogenic sources in smog chamber experiments. The exogenic PB-ROS content in the OOA from these emission sources was comparable to that in the ambient measurements. Our results imply that SOA from gaseous precursors of different anthropogenic emission sources is a crucial source of PB-ROS and should be additionally considered in toxicological and epidemiological studies in an adequate way besides primary emissions. The importance of PB-ROS may be connected to the seasonal trends in health effects of PM reported by epidemiological studies, with elevated incidences of adverse effects in warmer seasons, which are accompanied by more intense atmospheric oxidation processes.



# 1 Introduction

Air pollution has been shown to adversely impact human health through cardiovascular and respiratory disorders, which may ultimately lead to a reduction in life expectancy (WHO, 2013a; Pope et al., 2009; Salvi, 2007; Guarnieri and Balmes, 2014; Du et al., 2016; Cohen, 2000; Johnson, 2004). Epidemiological estimates suggest more than four million premature deaths per year due to air pollution, with a significant fraction attributed to China and India (Cohen et al., 2017; Lelieveld, 2017). Such analysis is based on relations between total particulate air pollution and health effects to provide exposure-response functions. A significant correlation between fine particle mass concentration and excess mortality was reported in the nineties in the "Harvard Six-City" study (Dockery et al., 1993) and consistently confirmed later on (e.g., Laden et al., 2006; (Lepeule et al., 2012; Beelen et al., 2014).

Since the composition of the atmospheric particulate matter (PM) depends on the sources and their chemical transformation in the atmosphere, an important question is which PM constituents are responsible for the adverse human health effects. Sulfate, organic carbon (OC) and transition metals have been shown to be more prominently associated with adverse health outcomes than other pollutants (Lippmann et al., 2013; Adams et al., 2015; Vedal et al., 2013; Burnett et al., 2000), even though there is no toxicological evidence supporting a causal role (WHO, 2013b). It is generally hypothesized that the adverse health effects caused by PM largely derive from oxidative stress induced by reactive oxygen species (ROS). The mechanism by which oxidative stress can be initiated is either through direct delivery of oxidants contained in PM (exogenous ROS) or by formation of endogenous ROS through introduction of redox-active compounds able to generate ROS in lung cells or by mitochondria as a response to particle exposure.

Different acellular assays have been explored to characterize the oxidant activity of PM. For instance, the 2′-7′-dichlorofluorescin (DCFH) and the p-hydroxyphenylacetic acid (POHPAA) assay respond to a range of organic peroxides (Venkatachari and Hopke, 2008; Wang et al., 2011a; King and Weber, 2013; Fuller et al., 2014; Zhou et al., 2018a; Hasson and Paulson, 2003) and are used to measure particle-bound ROS (PB-ROS). Similarly, the assay with 9,10-bis (phenylethynyl) anthracene nitroxide (BPEA-nit) measures the amount of PB-ROS including radicals and metals such as $Cu^+$ and $Fe^{2+}$, but not the ROS generated from Fenton chemistry (Miljevic et al., 2010; Hedayat et al., 2016). On the other hand, the dithiothreitol (DTT) assay is strongly sensitive to redox active components, i.e., transition metals and quinones and is considered as a good estimator of ROS generation in lung cells (Cho et al., 2005; Verma et al., 2012; Charrier and Anastasio, 2012; Fang et al., 2016; Weber et al., 2018), but is also sensitive to organic peroxides (Wang et al., 2018). The electron spin resonance (ESR) assay measures the capability of PM to induce hydroxyl radicals (Shi et al., 2003). Other assays measure the ability of PM to deplete biological antioxidants such as ascorbic acid (AA), glutathione and uric acid (Mudway et al., 2004; Fang et al., 2016). Since the different assays capture different fractions of the oxidant activity of PM, a direct comparison of measurements is challenging (Shiraiwa et al., 2017; Fang et al., 2016; Weber et al., 2018; Calas et al., 2017; Perrone et al., 2016; Yang et al., 2014; Janssen et al., 2015). Nonetheless, the ROS generation potential (referred to as oxidative potential, OP) determined using these assays was found to be more strongly associated with emergency department visits for airway and nasal inflammation, asthma, wheezing and congestive heart failure than $PM_{2.5}$ (PM with a particle diameter smaller than 2.5 μm) (Bates et al., 2015; Fang et al., 2016; Janssen et al., 2015).

More detailed analyses revealed associations between health effects and specific sources of particulate matter (Lippmann et al., 2013), such as wildfires, traffic, shipping, construction dust, metals sources, and coal and residual oil combustion



(Liu et al., 2017; Ostro et al., 2011; Lippmann Morton, 2013; Adam et al., 2015). To date, the emission sources governing ROS concentrations especially during haze episodes in developing and emerging countries are poorly constrained. Recent studies attributed the OP of ambient PM to different emission sources by applying source apportionment techniques like positive matrix factorization (PMF). Biomass burning and traffic contributed to the OP measured by DTT in most cases while contributions by secondary PM components like ammonium sulfate or the more-

oxidized fraction of OA were only found in some studies (Verma et al., 2015a; Fang et al., 2016; Bates et al., 2015; Weber et al., 2018). The OP measured by AA was found to correlate with Cu, secondary processes, and traffic (Weber et al., 2018; Fang et al., 2016), but not with biomass burning. The overwhelming majority of these studies are based on offline measurements with the inherent limitations related to filter sampling, including positive and negative artifacts and the loss of short-lived ROS during filter storage (Zhou et al., 2018a).

Here we systematically investigate the links between the chemical nature and sources of atmospheric aerosols and their PB-ROS content, by coupling aerosol mass spectrometry with online PB-ROS measurements and provide new insights into the sources of reactive oxidants in PM. We combine field data from two distinct environments, i.e. Beijing (China) and Bern (Switzerland), with results from comprehensive laboratory experiments, where we quantified the PB-ROS content in primary and secondary aerosols emitted from traffic, biomass burning, coal burning, and biogenic sources, the

most important sources of ambient organic aerosol. We show that PB-ROS, a large fraction of which is not accessible by offline measurements, is generated to a large extent in the atmosphere through oxidation processes.

## 2 Methods

### 2.1 Measurement campaigns

We combined the results from two field campaigns and three laboratory studies. One field campaign was conducted in

November 2014 at the Institute of Anatomy of the University of Bern (46°57'02.0"N 7°26'17.4"E), Switzerland, which is located in the city center but not directly affected by heavy traffic. The other field campaign took place in Beijing, China, from January to February 2015 at the National Center for Nanoscience and Technology (39°59'22.0"N 116°19'29.7"E, 20 m above ground), which is surrounded by university buildings and residential areas. For the laboratory studies two Atmospheric Chemistry Simulation Chambers of the Paul Scherrer Institute (PSI) were used, i.e., the 27-m$^3$ stationary

smog chamber (PSI-SSC) and the 7-m$^3$ mobile smog chamber (PSI-MSC). The PSI-SSC is a flexible bag made of fluorinated ethylene propylene (FEP) suspended in a temperature-controlled enclosure, where four xenon arc lamps (4 kW rated power, $1.55 \times 10^5$ lumen each, XBO 4000 W/HS, OSRAM) are used to simulate the solar light spectrum and to mimic natural photochemistry (Paulsen et al., 2005). Before each experiment, humidified pure air with high concentrations of ozone (> 2 ppm) was flushed into the chamber for at least 1 hour to clean the bag walls with lights on,

followed by flushing with dry, pure air for at least 10 h. The PSI-MSC is located inside a temperature-controlled housing flanked by 4 sets of 10 UV-lights (90–100 W, Cleo Performance, Philips) (Platt et al., 2013). In both chambers, OH was formed to a large degree from the photolysis of HONO and its concentration was evaluated by the decay of the marker compound d9-butanol (butanol-D9, 98%, Sigma-Aldrich) measured by a proton transfer reaction-mass spectrometer (PTR-MS, Ionicon Analytik GmbH) (Barmet et al., 2012). The gas- and particle-phase primary products that were

injected into the chambers were first thoroughly studied before the lights were switched on. Photochemical gas-phase reactions and formation of secondary organic aerosol (SOA) were then initiated by turning on the lights in the smog chambers. The following precursor emission sources were tested in this study:





**Wood combustion emissions:** In total, we investigated eight combustion devices with different technologies and combustion conditions, including a two-stage combustion updraft pellet boiler, a moving grate boiler equipped with an electrostatic precipitator, a pellet stove, a two-stage combustion downdraft log wood boiler, two advanced two-stage combustion log wood stoves, and two conventional single-stage combustion log wood stoves. Primary emission measurements from all these devices are reported here. For the secondary wood burning emissions, only the experiments conducted with one of the conventional single-stage combustion log wood stove and aged in the PSI-MSC are presented here, as it is more representative for the genuine atmospheric aging (with an OH exposure of $(2.6 - 4.8) \times 10^7$ molec cm$^{-3}$ h), compared to the emissions from the other burning devices that were aged in the potential aerosol mass (PAM) chamber (with an OH exposure of $(1.1 - 2.0) \times 10^8$ molec cm$^{-3}$ h). More details of the experimental method as well as the evolution of PB-ROS formation during the aging can be found in our previous publication (Zhou et al., 2018b).

**Coal combustion emissions:** We combusted three types of Chinese bituminous coal and two types of Chinese anthracite coal, in a conventional Chinese household cook stove. For each experiment, 100-300 g coal was pre-heated first by a hot honeycomb and anthracite coal until the temperature rose up to ~ 600 $^o$C, and then emissions were sampled, diluted 100-fold and injected into the PSI-MSC. One exemplary experiment showing the evolution of ROS and OA formation during the aging of the bituminous coal combustion emissions is presented in Fig. S1. Aging resulted in a strong increase of the total OA accompanied by formation of ROS reaching 0.6 nmol μg$^{-1}$ after 2 hours.

**α-pinene:** α-pinene (40 ppbv) was injected into the PSI-SSC containing O$_3$ (~ 500 ppbv) resulting in rapid formation of SOA. More details about the experimental procedure can be found elsewhere (Krapf et al., 2016). As seen in Fig. S2, the injection of α-pinene resulted in a fast production of OA and ROS, where ROS decreased strongly with time when α-pinene had reacted away, similar to the observed decay of peroxides by Krapf et al. (2016).

### 2.2 Instrumentation

A suite of online instrumentation was deployed to characterize the chemical and physical properties of gas and particle phase emissions in all the campaigns mentioned above.

**Reactive oxygen species (ROS) analyzer:** An on-line ROS analyzer based on the 2',7'-dichlorofluorescin (DCFH) assay was used for the quantification of the water-soluble particle-bound components acting as ROS (PB-ROS). A detailed description of the ROS analyzer is given in our previous publication (Zhou et al., 2018a). Particles were collected using an aerosol collector, of which the main part is a mist chamber. Before the aerosol collector, a honeycomb charcoal denuder was installed in a stainless steel tube to remove interfering gas phase compounds. The oxygen-free ultra-pure water (OF-OPW) was continuously sprayed into the mist chamber and incorporated the collected aerosol particles. The sample extracts were then mixed with the working solution (the DCFH solution) for analysis. The PB-ROS concentration was measured continuously and was calculated as H$_2$O$_2$ equivalents. All tested organic peroxides showed a linear response to DCFH, but the sensitivity is different and depends on their stability and neighboring functional groups. Transition metals and quinones that induce redox cycling and are well measured by the DTT assay do not react or interfere with DCFH when present at typical ambient concentration levels. This was also observed with the DTT assay for SOA (Wang et al., 2018). Thus, we conclude that DCFH measures the capability of particle-borne compounds to act as reactive oxygen species rather than the potential of species to mediate ROS formation (Zhou et al. 2018a).

**Aerosol mass spectrometer (AMS):** A field-deployable high-resolution time-of-flight aerosol mass spectrometer (HR-ToF-AMS, Aerodyne Research Inc.)(DeCarlo et al., 2006) was used for the characterization of the non-refractory aerosol mass in Beijing. Ambient air was sampled through a critical orifice into a recently developed aerodynamic lens which



efficiently transmitted particles between 80 nm and up to at least 3 µm (Williams et al., 2013). Particles were flash-vaporized by impaction on a resistively heated surface (~ 600 °C) and ionized by electron ionization (70 eV). The mass-to-charge ratios ($m/z$) of the resulting fragments were determined by a ToF mass spectrometer. Data were analyzed with the ToF-AMS software SQUIRREL and PIKA. Data was not corrected for lens transmission efficiency. For ambient data, standard relative ionization efficiencies (RIE) were used for organics (RIE=1.4), nitrate (RIE=1.1) and chloride (RIE=1.3). RIE for sulfate and ammonium were experimentally determined to be 1.12 and 3.58, respectively. The composition-dependent collection efficiency (CDCE) was evaluated using the algorithm by Middlebrook et al. (2012). For laboratory experiments, RIE of 1.4, 1.3, 1.1, and 1.2 were used for organics, chloride, nitrate, and sulfate, respectively, and experimentally determined 3.83 for ammonium. In line with past laboratory biomass burning experiments, a CDCE of 1 was used in all wood burning tests.

**Aerosol chemical speciation monitor (ACSM):** A quadrupole aerosol chemical speciation monitor (Q-ACSM, Aerodyne Research Inc.) was employed to monitor the composition and mass concentration of non-refractory submicron PM in Bern (Ng et al., 2011b). The Q-ACSM is built upon the same sampling and detection technology as the AMS described above, but with reduced complexity (e.g. no particle size measurement), resolution and performance (Fröhlich et al., 2013). Data was not corrected for lens transmission efficiency. Standard RIE were used for organics (1.4), nitrate (1.1) and chloride (1.3). RIE for sulfate and ammonium were experimentally determined to be 0.58 and 4.6, respectively. The CDCE was corrected based on the methodology described by Middlebrook et al. (2012).

**Aethalometer:** Real-time measurement of the optical absorption ($b_{abs}$) was performed by an aethalometer at 7 different wavelengths. A "next generation" aethalometer was used (AE33, Magee Scientific, Berkeley, CA, USA), which performed an online correction for possible scattering artefacts of the measured $b_{abs}$ (Drinovec et al., 2015). The $b_{abs}$ was then converted to an equivalent black carbon (eBC) concentration using the nominal mass absorption cross section (MAC) value of 11.8 m$^2$ g$^{-1}$ (Zotter et al., 2017).

**Gas phase instrumentation:** In the wood combustion PAM chamber experiments, total volatile organic compounds (VOCs) and CH$_4$ were measured with a flame ionization detector (FID) with a non-methane cutter (model 109A, J.U.M Engineering), CO and NO with a non-dispersive infrared analyzer (Ultramat 23 Siemens), O$_2$ with a paramagnetic oxygen analyzer (Ultramat 23 Siemens) and CO$_2$ with a non-dispersive infrared (NDIR) analyzer (model LI-820, LI-COR®). In the smog chamber (SC) aging experiments CO, CO$_2$, and CH$_4$ were measured with a cavity ring-down spectrometer (G2401, Picarro, Inc.). In all experiments, the composition of non-methane VOCs was determined by a high resolution proton transfer reaction–mass spectrometer (HR-PTR-MS 8000, Ionicon Analytik GmbH).

**VACES:** A versatile aerosol concentration enrichment system (VACES) was used to enrich the Bern ambient aerosol to increase the ROS concentration above the instrument detection limit. More detailed information on the VACES can be found in Künzi et al. (2015). The aerosol sample was drawn into the VACES at a flow rate of ~ 100 L min$^{-1}$ into a tank with water heated up to ~ 30 °C. The water vapor saturated air stream was then cooled down in a condenser (-2 °C) where water droplets with diameter > 2 µm formed on the collected aerosol particles. The droplets were then enriched in concentration with a virtual impactor and dried by passing through a diffusion dryer. The concentration enrichment factor CE was calculated as:

$$CE = Q_{tot}/q_{min}(1 - WL) \times \eta_{vi} \tag{1}$$

where $Q_{tot}$ and $q_{min}$ are the intake and minor flows of the impactor, respectively, $\eta_{vi}$ the collection efficiency, and WL the fractional loss of the impactor (Sioutas et al., 1999).





### 2.3 Statistical analysis

**Positive matrix factorization (PMF)** & **Multilinear engine (ME-2)**: Source apportionment was performed on the organic AMS and ACSM data using PMF as implemented by the multilinear engine (ME-2; Paatero, 1997) and controlled via the interface SoFi coded in Igor Wavemetrics (Source Finder(Canonaco et al., 2013)). PMF was developed by Paatero and Tapper (Paatero and Tapper, 1993, 1994) as a receptor model to apportion the sources on the basis of observations (internal correlations) at the receptor site alone (Viana et al., 2008), as represented by Eq. 2.

$$x_{ij} = \sum_{k=1}^{p}(g_{ik} \times f_{kj}) + e_{ij} \tag{2}$$

Here $x_{ij}$, $g_{ik}$, $f_{kj}$ and $e_{ij}$ are matrix elements of the measurements, factor time series, factor profiles and residual matrices, respectively, where $f_{ik} \geq 0$, $g_{ik} \geq 0$. The indices $i$, $j$, $k$ correspond to time, $m/z$ and a discrete factor, respectively. $p$ denotes the number of factors in the PMF solution determined by the user. The factor profiles remain constant, while their contributions to the matrix elements of the measurements are allowed to vary with time (concentration time series). In PMF, an "objective function" Q as defined in Eq. 3 is iteratively minimized:

$$Q(E) = \sum_{i=1}^{m}\sum_{j=1}^{n}[\frac{e_{ij}}{s_{ij}}]^2 \tag{3}$$

where $e_{ij}$ is the residual in the $i^{\text{th}}$ variable measured in the $j^{\text{th}}$ sample and $s_{ij}$ represents the corresponding "uncertainty" of $e_{ij}$. The main theoretical limitations of PMF are: 1) Source interpretation is relatively subjective; 2) Inability to clearly separate covariant sources (Viana et al., 2008) with similar chemical composition, e.g. cooking and traffic.

ME-2 provides a more flexible framework for controlling the solutions of the factor analysis where it permits the imposition of explicit external constraints (Ramadan et al., 2003). Within the *a*-value approach, *a priori* information (factor profiles or factor time series shown in Fig. S3) of one or more sources can be used as an additional input into the model, using the scalar *a* to regulate the strength of the constraints:

$$f_{j,solution} = f_j \pm a \times f_j \tag{4}$$

$$g_{i,solution} = g_i \pm a \times g_i \tag{5}$$

Here $0 \leq a \leq 1$, and $f_j$, $g_i$ are the input profiles and the time series, which can either entirely or partially be constrained by $a$.

**Aethalometer model:** The Aethalometer-based source apportionment model developed by Sandradewi et al. (2008) was used to separate the contributions of traffic (TR) and wood burning (WB) to the measured eBC in Bern. The equation used for the model is:

$$b_{abs,total}(\lambda) = b_{abs,tr}(\lambda) + b_{abs,wb}(\lambda) \tag{6}$$

Using the power law of the spectral dependence of the absorption, the measured $b_{\text{abs}}$ at two different wavelengths, and Eq. 6, TR and WB can be apportioned in the following way:

$$\frac{b_{abs,TR}(\lambda_1)}{b_{abs,TR}(\lambda_2)} = (\frac{\lambda_1}{\lambda_2})^{-\alpha TR} \tag{7}$$

$$\frac{b_{abs,WB}(\lambda_1)}{b_{abs,WB}(\lambda_2)} = (\frac{\lambda_1}{\lambda_2})^{-\alpha WB} \tag{8}$$

$$b_{abs,WB}(\lambda_2) = \frac{b_{abs}\lambda_1 - b_{abs}\lambda_2 \times (\frac{\lambda_1}{\lambda_2})^{-\alpha TR}}{(\frac{\lambda_1}{\lambda_2})^{-\alpha WB} - \frac{\lambda_1}{\lambda_2}^{-\alpha TR}} \tag{9}$$

$$b_{abs,TR}(\lambda_2) = \frac{b_{abs}\lambda_1 - b_{abs}\lambda_2 \times (\frac{\lambda_1}{\lambda_2})^{-\alpha WB}}{(\frac{\lambda_1}{\lambda_2})^{-\alpha TR} - \frac{\lambda_1}{\lambda_2}^{-\alpha WB}} \tag{10}$$





In Eqs. 7-10, α is the absorption Ångström exponent for traffic and wood combustion emissions ($\alpha_{TR}$ and $\alpha_{WB}$, respectively), which have to be assumed a priori. Similar to Fröhlich et al. (2015) and Sciare et al. (2011), $\alpha_{TR}=1$ and $\alpha_{WB}= 2$ were used, and $b_{abs}$ ($\lambda_1$) and $b_{abs}$ ($\lambda_2$), were chosen at 470 nm and 950 nm respectively. The $b_{abs}$ values attributed to the two sources were then converted to eBC using nominal mass absorption cross section (MAC) values from the instrument manufacturer. In this study, the model was only applied to the Bern data, where the contributions of other

combustion emissions to eBC were negligible (Zotter et al., 2017), but not in Beijing, as the contribution of coal combustion is substantial there (Liu et al., 2016; Elser et al., 2016) and large uncertainties regarding the optical properties of eBC emitted from coal combustion still exist.

**Bootstrap tests:** The bootstrap method, which is an effective approach to partially explore the influence of abnormal experimental conditions (i.e., the haze and reference episodes in Beijing, the presence of outliers, etc.) is employed in this

study to: 1) calculate a Pearson correlation matrix of all the testing factors in Beijing and Bern; 2) test the sensitivity of the multiple linear regression model (MLRM) to the input data as described in the following section 3.3. This statistical method is based on the creation of replicate inputs perturbing the original data by resampling. This was done by randomly reorganizing the rows of the original time series, so that some rows of the original data were present several times, while other rows were removed. Both of the analyses 1) and 2) were obtained by running the data for 1000

bootstrap replicates.

### 2.4 Definitions

**ROS fraction**: In order to study the ROS formation during aging, the secondary ROS content ($ROS_{SOA}$) was introduced. It describes the number of moles of secondary ROS ($ROS_S$ = aged ROS–primary ROS) per mass of secondary organic aerosol (SOA) formed during aging and was calculated as:

$$ROS_{SOA} = \frac{ROS_S}{SOA} \qquad (11)$$

Secondary organic aerosol (SOA) and secondary ROS ($ROS_S$) were calculated by subtracting primary organic aerosol (POA) and primary ROS ($ROS_P$) from the total OA and aged ROS, respectively, assuming $ROS_P$ and POA to be lost to the chamber walls at the same rate as eBC but otherwise to remain constant during aging. Although both quantities may not be conserved, a decrease of both did abate their effect on the PB-ROS fraction. In the SC experiments, POA was

defined as the OA mass before lights on, while SOA was estimated as the difference between total OA and the time dependent POA mass accounting for particle wall loss. Wall loss rates for POA and SOA were assumed to be the same as that of the measured eBC.

$f_{44\text{-SOA}}$: To express the degree of oxygenation of SOA, the fraction of secondary $Org44$ (corresponding to $m/z$ 44) in SOA (represented as $f_{44\text{-SOA}}$) is introduced, which was calculated as:

$$f_{44-SOA} = \frac{Org_{44-SOA}}{SOA} \qquad (12)$$

where $Org_{44-SOA}$ is the difference of total $Org_{44}$ and primary $Org_{44}$ and using the same procedure as for the SOA calculation mentioned above.

### 3 Results

The measurements in the city of Bern were performed in November 2014 using a quadrupole aerosol chemical speciation

monitor (Q-ACSM), an aethalometer, and the online ROS analyzer. In Beijing a high-resolution time-of-flight aerosol





mass spectrometer (HR-ToF-AMS), an aethalometer, and the ROS analyzer were used. Visibility was determined by an Automatic Weather Station (MAWS201, Vaisala, Vantaa, Finland) configured with a visibility sensor (Vaisala Model PWD22). Haze periods were defined by a visibility of less than 10 km while time periods with a visibility above 10 km were termed reference days (China Meteorological Administration, 2010).

### 3.1 Bulk chemical composition

Fig. 1a shows the average contributions of organic aerosol (OA), sulfate ($SO_4$), nitrate ($NO_3$), ammonium ($NH_4$), chloride (Cl), and equivalent black carbon (eBC) to $PM_{2.5}$ in Beijing in January-February 2015 (time series shown in Fig. S4). During haze events, its sum, considered as a proxy for $PM_{2.5}$, exceeded occasionally values of 200 µg m$^{-3}$ (average ≈ 110 µg m$^{-3}$). On reference days the $PM_{2.5}$ concentration ranged from 4 to 140 µg m$^{-3}$ with an average value of 28 µg m$^{-3}$. OA dominated in both haze and reference days, with a contribution of 49 ± 10% and 54 ± 8% to the total mass, respectively. As seen from the average diurnal patterns (Fig. S5), OA, eBC and Cl increased during the night time of the haze days, while $NO_3$ exhibited a maximum in the afternoon during both haze and reference days. During reference days, OA is high during night time, and additionally showed a significant afternoon peak concurrent with $NO_3$, possibly indicating additional secondary OA formation.

The average concentrations of $PM_1$ in Bern were ~ 15 and ~ 4 times lower than in Beijing ($PM_{2.5}$) during haze and reference periods, respectively (Fig. 1, $PM_1 ≈ 0.8\ PM_{2.5}$ in Bern and Beijing). The relative contributions of the different chemical components in the aerosols from Bern were similar to those from Beijing, except for chloride which contributed much less to the total mass in Bern than in Beijing. This may be due to hydrogen chloride (HCl) and methyl chloride ($CH_3Cl$) emissions from combustion sources in Beijing, e.g. coal emissions, incinerators and other industrial sources (Huang et al., 2014; Elser et al., 2016; McCulloch et al., 1999; Yudovich and Ketris, 2006).

### 3.2 Reactive oxygen species & OA sources

PMF was applied to the organic aerosol mass spectra (shown in Fig. S3) acquired by the ACSM and the AMS to extract the contributions of the different sources (see Section 2.3). Advanced error analyses methodologies were deployed to assess the PMF model uncertainties (as shown in Elser et al. (2016)). In Beijing, the result representing our data best was a 4-factor solution with three unconstrained factors, including oxygenated OA (OOA), a combination of hydrocarbon-like OA from traffic and coal burning emissions (denoted HOA + CCOA), cooking OA (COA), and a constrained biomass burning OA (BBOA) factor. Several tests were performed in the attempt to separate the HOA and CCOA factors, including the constraint of the factor profiles obtained in Elser et al. (2016). However, none of the tested approaches provided a satisfactory separation of these two sources, which were therefore recombined in the final solution. The BBOA factor profile obtained in previous ambient measurements in China by Elser et al. (2016) was constrained varying the *a*-value between 0 and 0.3 in steps of 0.1. In Bern, the solution that represented our data best was a 5-factor solution, with the four unconstrained factors OOA, BBOA, HOA, COA and a constrained fifth factor that could not be attributed to any specific source, but is required to properly extract COA in Bern, and is referred to as unidentified factor (Fig. S6a). In order to extract a clean profile of the unidentified factor (to be constrained in the final solution), we studied the changes in the mass spectra of the unidentified factor for an increasing number of factors (Fig. S6b). Based on these changes we constrained the unidentified mass spectra obtained in the 9-, 10-, and 11-factor solutions with *a*-values between 0 and 1 in steps of 0.1.





The factors were identified using their mass spectral fingerprints, and their source assignments were confirmed based on the factor diurnal patterns and their correlation with corresponding marker time-series. The OOA profile was

characterized by a dominant peak at *m/z* 44 and was associated with aged emissions and secondary organic aerosol formation. BBOA showed high contributions of *m/z* 29, *m/z* 60 ($C_2H_4O_2^+$), and *m/z* 73 ($C_3H_5O_2^+$), associated with cellulose pyrolysis products generated during biomass burning such as levoglucosan (Elser et al., 2016). The HOA+CCOA profile exhibited the typical fragmentation pattern of saturated and unsaturated hydrocarbons (Ng et al., 2011a). The COA profile was characterized by high signals at *m/z* 55 ($C_3H_3O^+$) (Crippa et al., 2013).

To explore the correlations between the fine aerosol chemical species, OA sources, and PB-ROS (referred to as testing factors in the following), we calculated a Pearson correlation matrix from 1000 bootstrap tests of those testing factors in Beijing and Bern (see Section 2.3 and Fig. S7). The average diurnal patterns of the OA components during the measurement periods in Beijing and Bern are shown in Fig. S8 and the time series of PB-ROS and the different identified source contributions are presented in Fig. 2. Site specific observations are discussed in the following.

***Beijing.*** As shown in Fig. 2a, PB-ROS as well as all the other aerosol species exhibited a simultaneous increase during the haze periods. Most of the variables had a strong ($r \geq 0.7 \pm 0.04$) or moderate ($r \geq 0.5 \pm 0.05$) positive Pearson correlation (Fig. S7a), which may be due to the influence of meteorological factors (Chen et al., 2017). HOA+CCOA correlated with eBC ($r = 0.78 \pm 0.04$), a tracer for traffic and coal combustion emissions. HOA+CCOA strongly decreased during noon, most likely due to a combined effect of decreased vehicular emissions (e.g. heavy-duty diesel

vehicles are only allowed to operate from 2200 to 0600 LT) (Bureau, 2004; Lin et al., 2009) and an increase in the mixing height. COA showed clear peaks during lunch (from 1200 to 1300 LT) and dinner time (from 1800 to 2300 LT) in both haze and reference days, consistent with local emissions from cooking processes. During both episodes, BBOA contributed less than 10% to the total OA (8.9% and 6.6% during haze and reference episodes, respectively), with little diurnal variability. The secondary inorganic components of $PM_{2.5}$ ($SO_4$, $NO_3$, and $NH_4$, see Fig. S5) followed the same

trend as OOA, suggesting that these components were formed through similar atmospheric transformation pathways. The PB-ROS concentration closely followed the trend of OOA with the highest average concentrations occurring during the afternoon and early night.

***Bern.*** In Bern, the Pearson correlation coefficients showed similar features, but were generally weaker than those in Beijing (Fig. S7b). The PB-ROS concentrations had moderate to high correlations with OA, $SO_4$, Cl, BBOA, eBC and

OOA ($r \geq 0.5 \pm 0.04$), but weak ones with $NO_3$, $NH_4$, COA and HOA ($r \leq 0.5 \pm 0.04$). HOA increased during the day with a small peak during the morning rush hour (0800 LT), representing urban traffic behavior. COA was elevated from late morning to early afternoon and again in the evening, reflecting local cooking activities. BBOA showed a clear increase during night time, related to residential heating with wood, a well-known aerosol source in Bern (Zotter et al., 2014). We separated eBC into the fossil fuel ($eBC_{TR}$) and wood burning ($eBC_{WB}$) fractions according to the method

described by Sandradewi et al. (2008) and Zotter et al. (2017). As shown in Fig. S8b, $eBC_{WB}$ had the same diurnal trend as BBOA, characterized by higher night time values, while $eBC_{TR}$ showed the opposite trend rather consistent with HOA, confirming the source apportionment results. The secondary inorganic component $SO_4$ correlated with OOA similarly to Beijing (Fig. 2b). These secondary components dominated the PM burden, indicating the importance of atmospheric aging to the chemical composition of the aerosol. PB-ROS levels also followed OOA in Bern (Fig. S8b) with no

substantial diurnal trends, except for a slight increase during daytime.

It should be noted that these correlations are not necessarily causative relations. For example, the inorganic species $NO_3$, $SO_4$, $NH_4$ correlated with each other due to their photochemical production and ammonium salt formation. As we





measured the water soluble PB-ROS concentration in the aerosol phase, eBC is not considered as a direct contributor to PB-ROS due to its water-insoluble character but indirectly by catalyzing the formation of oxidants at its surface in the

particle. The correlation of PB-ROS with eBC may also be a coincidence caused by the superposition of the two opposite trends of $eBC_{WB}$ and $eBC_{TR}$. Also, our previous tests showed that inorganic components like $NO_3$ and $SO_4$ did not contribute to the PB-ROS signal in the DCFH assay (Zhou et al., 2018a). Thus, we conclude that the correlations of the secondary inorganics with PB-ROS are a coincidence due to the fact that both are influenced by the atmospheric aging.

**3.3 Source contributions to PB-ROS**

A multiple linear regression model (MLRM) was applied to identify the dependence of the PB-ROS concentration on the different organic aerosol (OA) sources. First we fitted the observed ROS data as the dependent variable to all different OA sources obtained from the source apportionment as the predictors (function 1). Secondly, to exclude any influence of the separation of the individual primary sources, we tested the contribution of the total primary emission sources (here combined as primary organic aerosol, POA = (HOA + CCOA) + COA + BBOA) to the ROS activity in Beijing by

running the MLRM with only two variables, POA and OOA (using function 2 in Table 1). To assess the uncertainties related to the modeled ROS content values, we varied the source parameters by using 40 and 315 PMF solutions for Beijing and Bern, respectively. In addition, in each of these 40 (315) MLRM fits we used one set of PB-ROS concentrations generated by randomly varying the observed PB-ROS within the measurement uncertainties determined by the instrument precision and the background subtraction. The outcome of each MLRM fit provides one set of the PB-

ROS content of the different OA components. Finally, we also performed 1000 bootstrap runs on the final PMF solution (the average PMF results of all the solutions) to test the sensitivity of the MLRM to the input data, especially in Beijing where model results may be potentially driven by the haze periods. The average regression coefficients obtained from MLRM associated with each independent variable are listed in Table 1. For a better overview, the overall distributions of these regression coefficients are also illustrated as boxplots in Fig. S9.

The measured water soluble PB-ROS time series were fairly well captured by the model at both locations (Fig. 3) , with 61% of the variability explained in Bern and 77% in Beijing (see Fig. S10 for the normalized frequency distributions of $R^2$). For Beijing, the unexplained variance could be related to the measurement uncertainties, while for Bern, the model did not explain the entire variance in the data (22% of the remaining variability could be related to uncertainty and 17% remained unexplained). This unexplained part may be related to the variability of the composition/precursors of the OA

factors in Bern.

For Bern, PMF attributed 57% to the three primary sources BBOA (24%), HOA (20%), COA (13%) and 36% to OOA, as shown in Fig. 4b during the whole measurement period (note that the remaining 7% was attributed to the unidentified factor). However, these OA fractions contributed differently to PB-ROS. OOA was a much higher contributor to the observed PB-ROS (52% of explained PB-ROS) followed by contributions of BBOA (24%) and HOA (19%), and a small

contribution by COA (5%). In Beijing, the contribution of OOA to total OA was lower with 25%. HOA+CCOA was by far the highest primary source (43%) followed by small contributions of COA (24%) and BBOA (8%) during the whole measurement period. Fig. 4a shows the contribution of these OA factors during haze and reference days separately. However, concerning ROS, OOA was the main contributor to PB-ROS while the contributions of the individual primary OA sources to PB-ROS activity were statistically not different from zero within our uncertainties, when these sources

were considered individually (Table 1: function 1) or lumped together (Table 1: function 2). We note that this retrieval is dependent on (i) the covariance between factors used as dependent variables and (ii) on the factor relative contributions



to the overall ROS activity. In Beijing the OA factors are highly covariant, all increasing during haze. By contrast, in Bern, two regimes were encountered during the measurement period, with the dominance of primary biomass smoke and secondary organic aerosols, during the first and second half of the campaign, respectively. Such variability facilitated the

extraction of the contribution of both primary and secondary sources to ROS activity. The mass spectral profiles also show a higher O:C ratio of OOA in Beijing compared to Bern while it is the opposite for BBOA (Figure S3). This might indicate that PMF could have attributed the most oxidized fraction of BBOA to OOA in Beijing. Indeed, other source attribution analyses (Elser et al., 2016; Xu et al., 2018) have obtained higher BBOA fractions of 9-20% and a higher O:C ratio of 0.32-0.39 for BBOA compared to 7±5% and O/C=0.22 in this study. In addition, the sources of POA are

different at both locations. In Beijing, the contribution of HOA+CCOA, which has a low PB-ROS activity, is much larger than in Bern. By contrast, BBOA, which is ROS active, constitutes a large fraction of POA in Bern.

To further evaluate the contribution of primary emissions to PB-ROS in Beijing, we performed additional MLRM runs where the ratio of (ROS/OA)SOA/(ROS/OA)POA was constrained between 8 and 18 as shown in the laboratory studies below (Fig. 5). Even though (ROS/OA)POA and (ROS/OA)SOA are highly variable in the laboratory studies, their ratio

is not. With this approach the contribution of POA to the overall ROS in Beijing was determined to range between 9 and 18%. We note that the constrained and unconstrained models exhibited a statistically similar goodness of fit. This shows that the OOA ROS content is not sensitive to these constraints and suggests that the contribution of POA to the total PB-ROS activity in Beijing does not exceed 18%. Furthermore, in Fig. 3a we show the potential contribution from primary sources by adding one standard deviation to the regression coefficients listed in Table 1 (function 1). This illustrates, that

any enhanced contribution of POA could not improve the model.

On average, OA and PB-ROS were driven by different general overall sources derived from PMF & ME-2 (Fig. 4). For example in Beijing, even though the primary OA sources contributed ~ 75% to the total OA, their lumped contribution to PB-ROS was much lower.

### 3.4 Comparison of the ROS content of OA from different sources

Fig. 5 compiles online measurements of the PB-ROS content in aerosols measured by the DCFH assay from various ambient and laboratory measurements performed in this study or taken from literature. Ambient measurements are presented as the ROS content of the total measured aerosol mass (PM$_{2.5}$ in Beijing and PM$_1$ in Bern, denoted as ROS$_{PMx}$) or the organic aerosol (OA) fraction alone (ROS$_{OA}$) as determined from the MLRM. Note that PM$_{2.5}$/PM$_1$ is about 1.1-1.3, thus the usage of PM$_1$ in Bern does not much influence a comparison of ROS$_{PMx}$ and ROS$_{OA}$ from PM$_{2.5}$

measurements. Our values for ROS$_{PM2.5}$ (0.07 ± 0.04, and 0.09 ± 0.06, for Beijing_Haze, and Beijing_Reference, respectively) are in line with other online measurements reported by Huang et al. (2016) for Beijing (0.12 ± 0.05 and 0.10 ± 0.05 nmol µg$^{-1}$ in winter 2014 and spring 2015, respectively), and the value of ROS$_{PM1}$ (0.13 ± 0.06 nmol µg$^{-1}$) for Bern is not significantly different from the values in China. Figure 5b indicates the PB-ROS contents attributed to different sources identified in Bern and Beijing, and Figure 5c summarizes the PB-ROS contents from our own

laboratory measurements of different emission sources (see Section 2.1), complemented by literature values, i.e., 2-stroke scooter emissions (2s_scooter (Platt et al., 2014)), as well as SOA from limonene and oleic acid oxidation (Gallimore et al., 2017; Fuller et al., 2014). The PB-ROS contents in primary emissions (ROS$_{POA}$) from 2-stroke scooters or wood and coal burning are about 4 to 25 times lower than those in the corresponding SOA samples (ROS$_{SOA}$, see Section 2.4). Results are consistent with the field measurements, where the ROS content in OOA is 2-50 times higher than that in

primary emission sources (see Fig. S9 and Table 1). Clearly, the PB-ROS content is enhanced by photochemical



reactions, although the enhancement depends on the emitted precursors. SOA from anthropogenic emissions (vehicular exhaust and coal and biomass burning) includes in general a higher ROS activity compared to biogenic SOA from monoterpene precursors. For example SOA from α-pinene, the most emitted monoterpene compound has a 2-10 times lower PB-ROS content than SOA from the anthropogenic sources. Thus, despite the widespread dominance of biogenic

SOA, especially during summer, SOA formed from anthropogenic precursors might still dominate the PB-ROS burden in the atmosphere. Previously we have shown that the degree of oxygenation and the OA loading do influence the PB-ROS content of aerosols in a specific source (biomass burning) (Zhou et al., 2018b). However, Figure S11 shows that the averaged PB-ROS content in SOA from different emission sources does not significantly depend on these parameters in the given range but is more driven by the nature of the precursor. As VOCs emitted from wood and coal combustion

contain a large fraction of aromatic hydrocarbons (Bruns et al., 2017; Klein et al., 2018), this might indicate that aromatic VOCs yield a more PB-ROS active aerosol compared to monoterpenes.

Overall, the $ROS_{BBOA}$ derived from the Bern data was in the same range as $ROS_{POA}$ of wood burning emissions obtained from the laboratory study. The contribution of HOA+CCOA to PB-ROS was found to be negligible for Beijing within the analysis uncertainties (see Table 1), while in laboratory experiments PB-ROS was observed in primary emissions

from bituminous coal but not from anthracite coal. The fairly low and variable PB-ROS content in primary emissions from laboratory studies may explain why we could not derive its contribution to the measured PB-ROS in Beijing. $ROS_{OOA}$ in both Beijing and Bern were about 2 to 4 times lower than $ROS_{SOA}$ of gasoline and wood burning emissions, but equal or higher than $ROS_{SOA}$ of coal burning, α-pinene, limonene and oleic acid. It is reasonable to assume that the ambient OOA was formed from a complex mixture of precursors from various sources like traffic, wood burning, coal

burning and biogenic gases and, therefore, the ambient $ROS_{SOA}$ is expected to lie somewhere in the range of these sources.

ROS content measured with the BPEAnit assay, which also determines PB-ROS, was in the range of 0.05 -0.4 nmol μg$^{-1}$ for diesel/bio-diesel POA showing an increasing trend with the oxygen content of the fuel (Stevanovic et al., 2013; Hedayat et al., 2016). This assay also measured a higher PB-ROS content (0.2 -4 nmol μg$^{-1}$) in primary aerosols from

wood combustion (Miljevic et al., 2010). These concentrations are similar to our values for the scooter and coal POA as well as BBOA. In addition, a correlation between the oxygen content of diesel/bio-diesel POA and PB-ROS could be established and it was also shown that ageing of those emissions led to higher PB-ROS content in agreement with our observations (Stevanovic et al., 2013; Pourkhesalian et al., 2015).

**4 Atmospheric implications**

OA, dominated by secondary aerosol, is the largest fraction of PM$_1$ mass worldwide. Using both field and laboratory measurements we provide compelling evidence that PB-ROS are predominantly secondary. We estimate that while less than 40% of the OA mass at the urban locations investigated is secondary, this fraction contributes more than 60% of the total PB-ROS. These findings are corroborated by chamber experiments, showing that atmospheric aging does not only enhance the OA aerosol mass but also results in the formation of 10 times more ROS active material (in nmol μg$^{-1}$)

compared to POA. We show that SOA from anthropogenic complex precursor mixtures are especially ROS active, implying that despite their lower overall contribution to the OA burden compared to biogenic SOA, anthropogenic emissions can be an important PB-ROS source. Indeed, epidemiological studies revealed seasonal trends in health effects of PM. Studies in the US, Western Europe, Japan, South Korea including up to 112 cities and time periods up to 13 years



showed a stronger effect of PM on mortality in spring, summer or fall compared to winter (Peng et al., 2005; Nawrot et al., 2007; Zanobetti and Schwartz, 2009; Franklin et al., 2008; Yi et al., 2010). Kim et al. (2015) reported the highest overall risk of $PM_{10}$ for non-accidental or cardiovascular mortality on extremely hot days. However, in China the effects were highest in summer and winter (Kim et al., 2017; Chen et al., 2013). All in all, except for locations affected by strong haze and specific sources of toxic components (e.g., PAHs and coal combustion in winter in China), a stronger association between PM and mortality was observed at different locations during hotter days, when SOA formation is the most active. Various experiments have also shown that pure SOA does induce adverse effects upon exposure of airway epithelial cells (Arashiro et al., 2016, 2018; Gaschen et al., 2010; Lin et al., 2016; Chowdury et al., 2018; Jiang et al., 2016). Similarly, cellular based ROS assays show a strong connection with pure SOA or its fraction in PM (Tuet et al., 2016; Tuet et al., 2017a; Tuet et al., 2017b; Saffari et al 2014). Further, Lakey et al. (2016) developed a model to simulate the response of air pollutants including PM on the production rates and concentrations of ROS in the epithelial lining fluid of the human respiratory tract. They found that the OH production rate from SOA in the lung could be as high as the $H_2O_2$ production rate from redox-active trace metals and quinones. Others reported that SOA dissolved in water exhibited a high OH radical or $H_2O_2$ generation under irradiation (Wang et al., 2011b; Lim, 2015; Badali K. M. , 2015) or a direct formation of OH under dark (Tong et al., 2016). The DCFH assay for exogenous ROS has so far not been applied as tracer for ROS activity and oxidation potential of PM to our knowledge. The most frequently used assay with DTT shows in some cases good associations with oxidized OA and high values in summer while in other studies better relations with biomass burning and traffic PM components as well as high winter values are observed (Fang et al., 2016; Verma et al. 2014, 2015; Bates et al., 2015). The DTT assay is sensitive to a broad range of components, peroxides and redox cycling species like transition metals and quinones but may suffer from negative interferences between metals and quinones (Wang et al., 2018). Thus, a combination of measurements might be needed to better assess relations between PM composition, oxidative potential and possible health effects. The DCFH assay responds to peroxides but not to quinones and transition metals and could thus be helpful to disentangle the various contributions in combination with a DTT assay. It is also known that particles do carry short and long-lived ROS (Zhou et al., 2018a; Fuller et al., 2014; Krapf et al.2016), whereby the former are mostly lost when filters are analyzed. A combination of different on-line and off-line methods appears to be more suited for an optimized understanding of the various short- and long-lived ROS species. Research on links between PB-ROS quantified by the DCFH assay and adverse biological effects of PM should also be envisaged.

**Acknowledgements**

This study was financially supported by the Swiss National Science Foundation (NRP 70 "Energy Turnaround", CR32I3-140851, CR32I3_166325, and 200021L_140590), the National Natural Science Foundation of China (NSFC) under Grant No. 91644219 and No. 21661132005, the National Key Research and Development Program of China (No. 2017YFC0212701), and the China Scholarship Council (CSC). T.N. and P.Z. acknowledge the Swiss Competence Center for Energy Research SCCER BIOSWEET of the Swiss Innovation Agency Innosuisse.

**Author contributions**




Study design (J.Z., J.D., U.B., M.G., R.-J.H., I.E.H., A.S.H.P.); experimental work, Bern ambient (J.Z., J.D., M.K., R.F., M.G.); experimental work, Beijing ambient (J.Z., M.E., J.D., R.-J.H., Q.Y.W., Y.W., Y.Z.); experimental work, wood burning (J.Z., D.B., G.S., P.Z., E.A.B., S.B.); experimental work, coal burning (J.Z., R.-J.H., S.M.P., H.N., G.S.); experimental work, α-pinene (J.Z., M.K., J.D.); data analysis, reactive oxygen species (J.Z.); data analysis, AMS data for
Beijing (M.E.); ACSM data, eBC source apportionment for Bern (R.F.); data analysis, source apportionment for Beijing and Bern (M.E., R.F.); data analysis, AMS for laboratory studies (G.S., D.B., S.M.P); writing of manuscript (J.Z.); data interpretation and editing of manuscript (J.Z., J.D., U.B., R.-J.H., I.E.H., A.S.H.P.); preparation of display items (J.Z., M.E.); literature data ambient/laboratory (J.Z., J.D.); comments and discussion on the manuscript (all).

**Competing interests**

The authors declare that they do not have any financial or non-financial interests in relation to the work described.

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



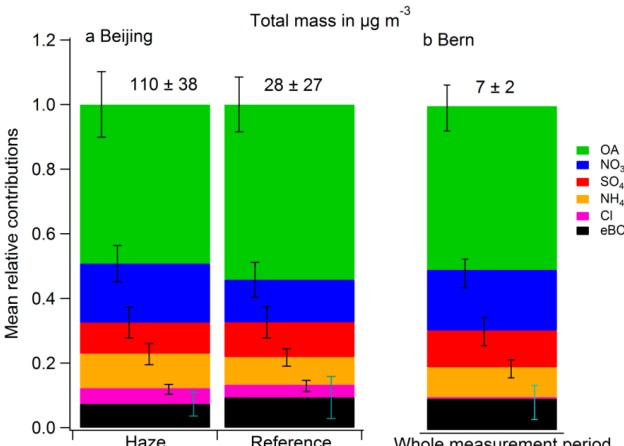

**Figure 1:** Total PM mass concentrations and relative contributions of non-refractory chemical components plus eBC in (**a**) Beijing (PM$_{2.5}$, January-February, 2015) and (**b**) Bern (PM$_1$, November, 2014).



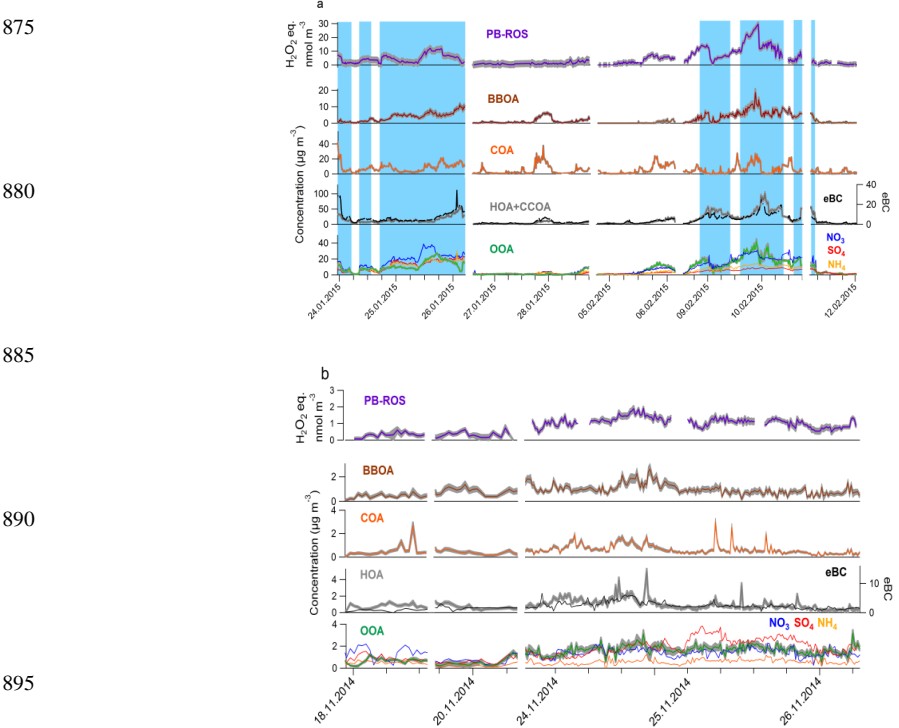




**Figure 2:** Concentrations of PB-ROS, OA components, and eBC during the measurement periods in (**a**) Beijing and (**b**) Bern. In Beijing, the periods highlighted with a blue background represent the haze periods (visibility < 10 km). The remaining periods were termed reference periods (visibility ≥ 10 km). The gray bands indicate $1\sigma$ errors of the PB-ROS measurements and of our best estimates of the factors.




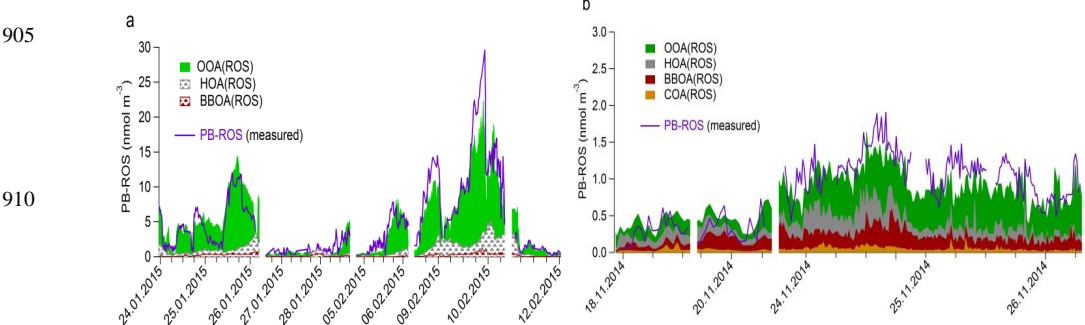

**Figure 3:** Comparison of measured (purple line) and simulated PB-ROS concentrations in (**a**) Beijing and (**b**) Bern, where HOA(ROS), COA(ROS), BBOA(ROS) and OOA(ROS) denote the corresponding source contributions to the simulated PB-ROS activity. Note that in Beijing, the HOA(ROS) and BBOA(ROS) represent the potential contribution of the HOA and BBOA contributions to PB-ROS, by adding one standard deviation to the regression coefficients listed in Table 1 (function 1).

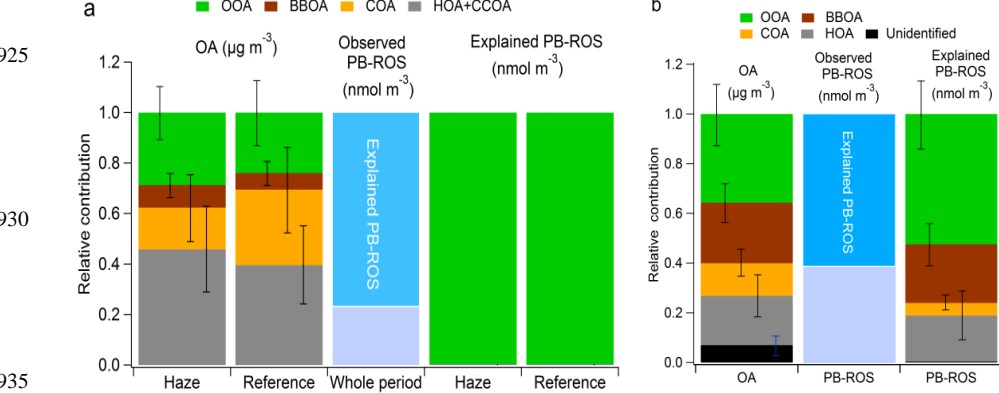

**Figure 4:** Average relative contributions of the OA sources to the observed total OA and to the explained PB-ROS during the measurement period in (**a**) Beijing and (**b**) Bern. Data in Beijing is separated for haze and reference periods. The error bars represent the standard deviation of the mean of the whole measurement period.





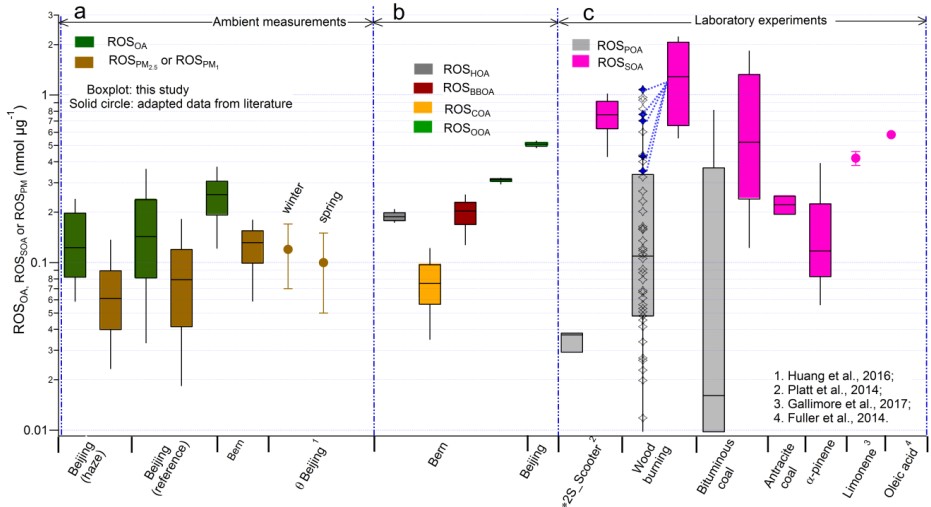

**Figure 5** Comparison of the ROS content in aerosols from different sources (listed on the x-axis) as measured by the DCFH-assay. (**a**) PB-ROS content in OA ($ROS_{OA}$) and PM ($ROS_{PM2.5}$ for Beijing and $ROS_{PM1}$ for Bern) at ambient sites. For comparison, the ROS content for winter and spring in Beijing (represented as θ Beijing) is calculated from the ROS concentrations reported in the literature in equivalent nmol $H_2O_2$ per $m^3$ air and then normalized to $PM_{2.5}$ concentrations obtained from the U.S. Department of State (Embassy); (**b**) PB-ROS content of individual source factors in Bern and Beijing; (**c**) PB-ROS contents in primary ($ROS_{POA}$) and secondary organic aerosol ($ROS_{SOA}$) from wood and coal burning), in SOA from α-pinene (this study, as well as literature data for 2-stroke scooters (2s_Scooter)(Platt et al., 2014), and for SOA from limonene (Gallimore et al., 2017) and oleic acid (Fuller et al., 2014). The blue points and the blue dotted lines in the wood burning experiment are used to identify the corresponding $ROS_{POA}$ of the listed $ROS_{SOA}$. *For the $ROS_{POA}$ of the 2S_Scooters (gasoline emissions) we take only the points right before lights on from the original data as earlier data points seemed still be influenced by incomplete mixing of emissions in the chamber.

**Table 1:** The average regression coefficients and their standard deviation (1 σ) for MLRM with function 1 and 2.

| Function 1 | | PB-ROS = $a$ × (HOA or HOA+CCOA)+ $b$ × COA + $c$ × BBOA + $d$ × OOA+ intercept | | | | |
|---|---|---|---|---|---|---|
| Parameter | | $a$ (HOA) | $b$ (COA) | $c$ (BBOA) | $d$ (OOA) | intercept |
| | 40 fits | 0.01±0.04 | -0.01±0.01 | -0.04±0.11 | 0.51±0.02 | 0.07±2.00 |
| Beijing | bootstraps | 0.01±0.01 | 0.003±0.009 | -0.01±0.056 | 0.53±0.19 | -0.00±0.00 |
| Bern | | $a$ (HOA+CCOA) | $b$ (COA) | $c$ (BBOA) | $d$ (OOA) | intercept |
| | 315 fits | 0.19±0.02 | 0.08±0.03 | 0.20±0.06 | 0.31±0.02 | 0.11±0.12 |
| | bootstraps | 0.21±0.03 | 0.08±0.05 | 0.18±0.04 | 0.34±0.03 | 0.00±0.00 |
| Function 2 | | PB-ROS = $e$ × (HOA+CCOA + COA + BBOA )+ $f$× OOA+ intercept | | | | |
| Beijing | | $e$ (HOA+CCOA+COA+BBOA) | | | $f$ (OOA) | intercept |
| | 40 fits | 0.004±0.02 | | | 0.51±0.02 | 0.7±2.00 |

± represents one standard deviation (1 $\sigma$) of the average coefficients of all the MLRM fittings (see text for details).