# Peer review of "Predominance of Secondary Organic Aerosol to Particle-bound Reactive Oxygen Species Activity in Fine Ambient Aerosol"

_Atmospheric Chemistry and Physics, 2019_

## Referee Comment (RC1) · Zoran Ristovski (Referee) · 18 Jul 2019

Review of acp-2019-190

The manuscript by Zhou et al. presents results from real time PB-ROS measurements, conducted with a DCFH probe, both in field as well as in smog chamber studies. The authors show that the OOA is the main contributor to the to the PB-ROS. What is further of interest is that the ROS measurements from chamber studies show a similar level of PB-ROS to the ambient measurements and that the oxidation of anthropogenic gaseous precursors is a significant contributor to the PB-ROS and will dominate the PB-ROS in urban environments. This is an important finding and will significantly con-

tribute to our understanding of the PM health effects. Further the manuscript clearly points to a need to use different assays "to better assess relations between PM composition, oxidative potential and possible health effects."

Overall this is a very well written manuscript and should be accepted after some minor improvements. The only more major comment is that the authors do not really have a long time data set. On the other side taking into account that real time PB-ROS measurements are fairly complex and still in development this is not surprising.

Questions/Suggestions for improvement:

p.4 Instrumentation. Although a detailed description of the on-line PB-ROS instrument was given it would be beneficial to mention few more characteristic of the instrument. For example what was the flow rate of the instrument and what was the time resolution. Also if you have some estimates on the sensitivity of the instrument it would give more confidence in the observed data. I understand that the sensitivity cannot be expressed in units normalized per PM mass but expressing this normalized per volume of air sampled (nmols/m3) would be possible.

p.4, l.139 "Before the aerosol collector, a honeycomb charcoal denuder was installed in a stainless steel tube to remove interfering gas phase compounds." Do you have any proof of the effectiveness of the denuder in removing the gas phase compounds? It has been shown that the gas phase ROS can significantly contribute to the total ROS (particle + gas phase) therefore if you do not efficiently remove the gas phase your PM-ROS measurements could be biased.

p.5, l.181 VACES. Could the condensation of water on the particles followed by evaporation, that we have in the VACES, have any influence on the PB-ROS concentrations. A comment on this would be beneficial.

p.11, l.386 "Furthermore, in Fig. 3a we show the potential contribution from primary sources by adding one standard deviation to the regression coefficients listed in Table

1 (function 1)." I cant see this in figure 3a. Is this missing from figure 3a or I did not understand the statement?

p.11, l.405 "Our values for ROSPM2.5 (0.07 $\pm$ 0.04)..." what are the units?

P12, l.422 "However, Figure S11 shows that the averaged PB-ROS content in SOA from different emission sources does not significantly depend on these parameters..." To which of the 2 parameters, degree of oxygenation and/or the OA loading, are you referring to or both? It has been shown in the past both by Stevanovic et al (2013), Hedayet et al (2016) as well as your measurements Zhou et al (2018) (Figure 4) that the degree of oxidation expressed through f44 has an influence on the PB-ROS. Are you claiming that this is the case only for one type of source i.e. primary diesel or wood combustion? If yes please be more specific and argument this.

---

## Referee Comment (RC2) · Anonymous Referee #3 · 14 Aug 2019

This manuscript discusses interesting and significant work that relates to real time PB-ROS measurements for particles collected from field campaigns and laboratory studies. The authors apportion the ambient PB-ROS to different OA sources and the results are further supported by laboratory chamber studies. The findings are important, and the manuscript is well written. I recommend that it can be published following some revisions.

1. Both field campaigns were not conducted in warm seasons, it may be too early to connect PB-ROS to elevated incidence of adverse effects in warmer seasons.

2. OA sources from PMF is based on online ACSM or AMS data. But the PB-ROS

measurement is from water soluble fraction of the aerosol. Should the solubility of each OA factor be considered when attribute their contribution to the PB-ROS?

3. Line 145: "Transition metals and quinones that induce redox cycling and are well measured by the DTT assay do not react or interfere with DCFH when present at typical ambient concentration levels." Did the author do some tests and conclude it? If yes, please show which transition metals and quinones did the author test? Different quinone or transition metal species can show varying sensitivities. This has been found for DTT assay (Charrier et al., ACP 2012). Also, it is known that transition metals and quinones in aqueous solution can form H2O2, which is very sensitive to DCFH. Since the authors did not see any DCF signal, does it mean the formation of H2O2 is too low or due to the mixing time of transition metals and quinones with water is too short?

Minor comment:

1. eBCWB and eBCTR are not defined in the manuscript

2. Fig S7a, use "OA" in both x and y axis labels.

---

## Author Comment (AC1) · 17 Sep 2019

We would like to thank the referees for their positive feedback and the constructive comments, which we will address point-by-point below. The reviewer's comments are in black, our answers are reported in red and the modifications to the manuscript are marked in yellow.

**Zoran Ristovski (Referee)**

z.ristovski@qut.edu.au

The manuscript by Zhou et al. presents results from real time PB-ROS measurements, conducted with a DCFH probe, both in field as well as in smog chamber studies. The authors show that the OOA is the main contributor to the to the PB-ROS. What is further of interest is that the ROS measurements from chamber studies show a similar level of PB-ROS to the ambient measurements and that the oxidation of anthropogenic gaseous precursors is a significant contributor to the PB-ROS and will dominate the PB-ROS in urban environments. This is an important finding and will significantly contribute to our understanding of the PM health effects. Further the manuscript clearly points to a need to use different assays "to better assess relations between PM composition, oxidative potential and possible health effects."

Overall this is a very well written manuscript and should be accepted after some minor improvements. The only more major comment is that the authors do not really have a long time data set. On the other side taking into account that real time PB-ROS measurements are fairly complex and still in development this is not surprising.

**Questions/Suggestions for improvement:**

p.4 Instrumentation. Although a detailed description of the on-line PB-ROS instrument was given it would be beneficial to mention few more characteristic of the instrument. For example what was the flow rate of the instrument and what was the time resolution. Also if you have some estimates on the sensitivity of the instrument it would give more confidence in the observed data. I understand that the sensitivity cannot be expressed in units normalized per PM mass but expressing this normalized per volume of air sampled (nmols/m3) would be possible.

A detailed characterisation of the on-line PB-ROS instrument was reported in our previous instrument paper (Zhou et al., 2018a). We agree that it is helpful for the reader to add some instrument characteristics to the instrument description here. We added the flow rates and the time resolution of the instrument in section 2.2: P. 4, line 139 "Particles were collected at a flow rate of ~1.7 L min-1, using..." and on P. 4, line 143 "The PB-ROS concentration was measured continuously with a response time of 8 minutes (rise time of signal from 10 to 90% of full signal) and...".

Regarding the sensitivity of the instrument we assume the referee refers to the instrument limit of detection (LOD). With respect to  $H_2O_2$  a LOD of 2 nmol m-3 of sampled air was determined. We added this information on P4, line 143-144 "An instrument limit of detection of 2 nmol m-3 of sampled air was determined".

p.4, I.139 "Before the aerosol collector, a honeycomb charcoal denuder was installed in a stainless steel tube to remove interfering gas phase compounds." Do you have any proof of the effectiveness of the denuder in removing the gas phase compounds? It has been shown that the gas phase ROS can significantly contribute to the total ROS (particle + gas phase) therefore if you do not efficiently remove the gas phase your PM-ROS measurements could be biased.

We performed several gas-phase interference tests. In principle, at the applied sample flow rate, 99% of the trace gases should get removed by the denuder. Specifically, we assessed

the removal efficiency of the denuder with respect to the most abundant oxidizing trace gases  $O_3$  and  $NO_2$ . After exposing the denuder to 464 ppb ozone for ~5 h, no increase in the background signal was observed (Table 2 in Zhou et al., 2018a). An amount of 500 ppb  $NO_2$  showed no increase in the background signal even without the denuder. The results in Table 2 (Zhou et al., 2018a) indicate that a newly regenerated denuder completely removes  $O_3$ , making the denuder suitable for both smog chamber (usually ~5 h aging per experiment) and ambient measurements (1 day replacement interval). Based on these results we assume that gaseous  $H_2O_2$  is also completely removed. Further, we regularly checked the ROS blank by measuring particle-free air by switching a three-port valve and sampling through a particle filter (disposable filter units, Balston, UK) installed in another line. We clarified this at the end of the instrument description: page 4, lines 149-151 "Before the aerosol collector, a honeycomb charcoal denuder was installed in a stainless steel tube to remove interfering gas phase compounds. Tests revealed no interference from  $NO_2$  nor ozone up to 2300 ppb h. Denuders were regenerated every day."

p.5, I.181 VACES. Could the condensation of water on the particles followed by evaporation, that we have in the VACES, have any influence on the PB-ROS concentrations. A comment on this would be beneficial.

We investigated this during the measurements in Bern. We added now a comment on this in page 6, line 193-197: A potential influence of the enrichment process on the PB-ROS concentrations was investigated during the Bern ambient air measurements. We compared off-line PB-ROS measurements using filters collected before and after the VACES with the on-line PB-ROS measurements, which were always done after the VACES. No differences of ROS concentrations were observed between filters taken before and after VACES compared with the online measurements (Zhou et al., 2018a).

p.11, I.386 "Furthermore, in Fig. 3a we show the potential contribution from primary sources by adding one standard deviation to the regression coefficients listed in Table 1 (function 1)." I cant see this in figure 3a. Is this missing from figure 3a or I did not understand the statement?

The regression coefficients of the primary sources listed in Table 1 are statistically not different from zero. Therefore, in Fig 3a we show the potential contribution from these primary sources by using the upper limit of the regression coefficient [regression coefficients + one standard deviation]. We revised the sentence for clarity as follows on page 11, line 399-401: "Furthermore, in Fig. 3a we show the potential contribution from primary sources, which is calculated from the upper limit of their regression coefficients (regression coefficient + one standard deviation) listed in Table 1 (function 1)".

p.11, I.405 "Our values for ROSPM2.5 (0.07 \_ 0.04)..." what are the units?

It is  $nmol \mu g^{-1}$ , we added this unit on page 11, line 413.

P12, I.422 "However, Figure S11 shows that the averaged PB-ROS content in SOA from different emission sources does not significantly depend on these parameters..." To which of the 2 parameters, degree of oxygenation and/or the OA loading, are you referring to or both? It has been shown in the past both by Stevanovic et al (2013), Hedayet et al (2016) as well as your measurements Zhou et al (2018) (Figure 4) that the degree of oxidation expressed through *f*44 has an influence on the PB-ROS. Are you claiming that this is the case only for one type of source i.e. primary diesel or wood combustion? If yes please be more specific and argument this.

We apologize that this is not clear. We agree with the reviewer that the PB-ROS content in SOA from different sources depends on both, degree of oxygenation and OA loading. This dependence is reflected in the standard deviation of each source in Figure S11. However, in Figure S11 we show that the variation of PB-ROS content by these two parameters within a source is smaller than the variation of the average PB-ROS content between different sources or precursors. To make it clear, we have changed this sentence to "However, Figure S11 shows that the variation of PB-ROS in SOA within a specific source is smaller than the differences between the sources and precursors. This indicates that the PB-ROS content is more driven by the nature of the precursor" in page 12, line 430-432.

**Anonymous Referee #3**

This manuscript discusses interesting and significant work that relates to real time PBROS measurements for particles collected from field campaigns and laboratory studies. The authors apportion the ambient PB-ROS to different OA sources and the results are further supported by laboratory chamber studies. The findings are important, and the manuscript is well written. I recommend that it can be published following some revisions.

1. Both field campaigns were not conducted in warm seasons, it may be too early to connect PB-ROS to elevated incidence of adverse effects in warmer seasons.

Indeed, both field campaigns were performed in winter time. Nevertheless, we find OOA and not POA to be the predominant PB-ROS contributor. OOA is formed from atmospheric oxidation in winter as well as in summer. We acknowledge that the relative contribution of emission sources will be different between summer and winter. However, our laboratory experiments of different emission sources all show that atmospheric aging considerably increases PB-ROS. Thus it is highly probable that OOA will be the predominant PB-ROS source also in summer. We do not claim a direct connection between PB-ROS and elevated incidence of adverse effects in warmer seasons. We only venture a guess in that direction, (e.g. in the abstract: "The importance of PB-ROS *may* be connected to the seasonal trends in health effects of PM reported by epidemiological studies, with elevated incidences of adverse effects in warmer seasons, which are accompanied by more intense atmospheric oxidation processes"), which we think is highly possible and should be looked at in more detail.

2. OA sources from PMF is based on online ACSM or AMS data. But the PB-ROS measurement is from water soluble fraction of the aerosol. Should the solubility of each OA factor be considered when attribute their contribution to the PB-ROS?

Indeed, we measure only the water soluble fraction of PB-ROS. With our method we cannot measure PB-ROS of the water insoluble fraction. Therefore, we report the contribution of each OA factor to the amount of water soluble PB-ROS. This is stated in the paper. Scaling with the solubility of each factor would yield the specific PB-ROS per water soluble fraction of an OA factor. Although this may be an interesting number, it is a different physico-chemical quantity.

To better clarify it from the beginning we specified it better in the abstract by writing "watersoluble PB-ROS".

3. Line 145: "Transition metals and quinones that induce redox cycling and are well measured by the DTT assay do not react or interfere with DCFH when present at typical ambient concentration levels." Did the author do some tests and conclude it? If yes, please show which transition metals and quinones did the author test? Different quinone or transition metal species can show varying sensitivities. This has been found for DTT assay

(Charrier et al., ACP 2012). Also, it is known that transition metals and quinones in aqueous solution can form H2O2, which is very sensitive to DCFH. Since the authors did not see any DCF signal, does it mean the formation of H2O2 is too low or due to the mixing time of transition metals and quinones with water is too short?

Yes, such tests have been done and are reported in our previous paper Zhou et al. 2018a. We investigated iron, one of the most abundant transition metals in the aerosol, and anthraquinone for their response in the DCFH assay. Furthermore, we tested whether the complex matrix of ambient particles, which include different forms of iron together with other metals, interferes with the PB-ROS measurements. In this second set of experiments, ambient filter samples were extracted and cross tested with  $H_2O_2$ . Results indicated that at concentrations relevant for the ambient atmosphere the complex matrix of ambient particles has no influence on the PB-ROS signals (Zhou et al., 2018). This further demonstrated that at typical ambient concentration levels, the transition metals and quinones that induce redox cycling do not react or interfere with DCFH. In addition, at ambient levels of metals and quinones their concentrations in the sample solution would be too low and the reaction time too short to produce measurable amounts of  $H_2O_2$ .

We have specified the tested transition metals and quinone in page 4, line 146 in the modified manuscript.

"Transition metals (i.e., Fe) and quinones (i.e., anthraquinone) that induce redox cycling and are well measured by the DTT assay do not react or interfere with DCFH when present at typical ambient concentration levels."

Minor comment:

1. eBCWB and eBCTR are not defined in the manuscript We actually defined them on p. 9, line 334.

2. Fig S7a, use "OA" in both x and y axis labels.

We modified it in the new version.

**Addendum**

After a re-analysis of the coal combustion experiments we noticed that SOA formation from anthracite coal cannot be unambiguously separated from the organic aerosol formation in back ground experiments. Therefore, we decided to eliminate this information in Figure 5 and indicated this in the text on page 12, line 436:

The contribution of HOA+CCOA to PB-ROS was found to be negligible for Beijing within the analysis uncertainties (see Table 1), while in laboratory experiments PB-ROS was observed in primary and secondary emissions from bituminous coal but not from anthracite coal.